# AdaCache: Adaptive Caching and Context Augmentation for Efficient LLM Serving

**Zihao Zeng[1,2]\*, Siyi Li[1], Xinyu Yan[1,2], Lei Xiao[3], Wei Yang Bryan Lim[1]**
[1]Nanyang Technological University
[2]Alibaba-NTU Global e-Sustainability CorpLab (ANGEL)
[3]Alibaba Group
{zihao.zeng, bryan.limwy}@ntu.edu.sg

## Abstract

Retrieval-Augmented Generation (RAG) significantly enhances Large Language Models by integrating external knowledge sources, but at the cost of substantial computational overhead from extended input sequences. Current RAG systems exhibit two fundamental inefficiencies: redundant processing of frequently retrieved text chunks across multiple queries, and uniform deep retrieval that over-provisions context regardless of query complexity. We present AdaCache, an adaptive caching framework that addresses these limitations through dual optimization strategies. First, we introduce a cache-aware partial recomputation mechanism that profiles attention patterns to construct selective cache variants, enabling flexible reuse while preserving cross-chunk dependencies. Second, we develop adaptive context augmentation that dynamically determines optimal retrieval depth via lightweight confidence estimation, avoiding unnecessary overhead on simple queries. Comprehensive experiments across diverse datasets and LLMs demonstrate that AdaCache delivers substantial improvements in Time-To-First-Token compared to state-of-the-art RAG caching systems, while preserving generation quality.

## 1 Introduction

Large Language Models (LLMs) have become ubiquitous across diverse applications, from conversational chatbots and personal assistants to specialized systems handling question answering, document summarization, and machine translation (Achiam et al., 2023; Hurst et al., 2024; Guo et al., 2025; Yang et al., 2025). Despite their impressive capabilities, LLMs suffer from hallucination issues and knowledge limitations, particularly when dealing with domain-specific or up-to-date information. Retrieval-augmented generation (RAG) (Ram et al., 2023; Siriwardhana et al., 2023; Jiang et al., 2023) has emerged as a powerful paradigm to bridge this gap. By incorporating external knowledge bases, such as Wikipedia (Cohere, 2023) or domain-specific corpora, it retrieves relevant contextual information to enrich user queries. This approach has demonstrated remarkable success in improving generation quality, while enabling general-purpose LLMs to tackle specialized domain problems without costly fine-tuning.

Despite these benefits, RAG introduces significant system-level challenges. The injection of retrieved text chunks substantially increases the length of input prompts, leading to higher computation and memory requirements during the LLM inference. For instance, while a raw user query typically contains fewer than 200 tokens, augmenting it with retrieved context can push the sequence length beyond 2,000 tokens, leading to more than a $10\times$ increase in computational and memory overhead. This dramatic expansion significantly degrades Time-To-First-Token (TTFT) and system throughput, ultimately compromising user experience. The key objective is to achieve the best of both worlds: harnessing RAG's quality improvements while preserving computational efficiency.

Our observation reveals two major inefficiencies in current RAG systems. The first is cross-query context overlap, where identical text chunks from the external knowledge base are repeatedly re-

---

\*Corresponding author.

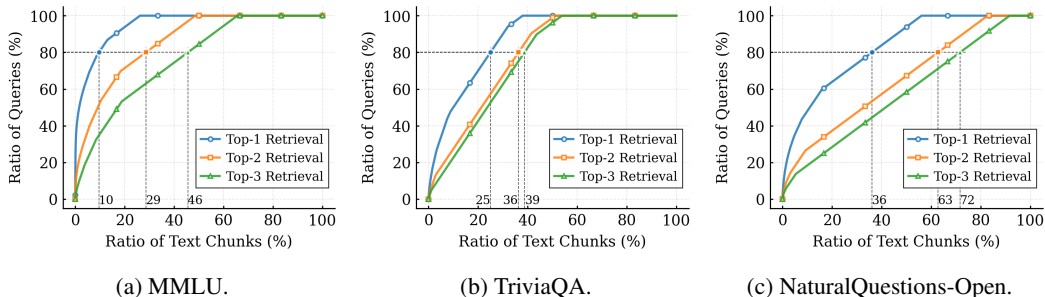

(a) MMLU.     (b) TriviaQA.     (c) NaturalQuestions-Open.

Figure 1: Retrieval pattern on different datasets.

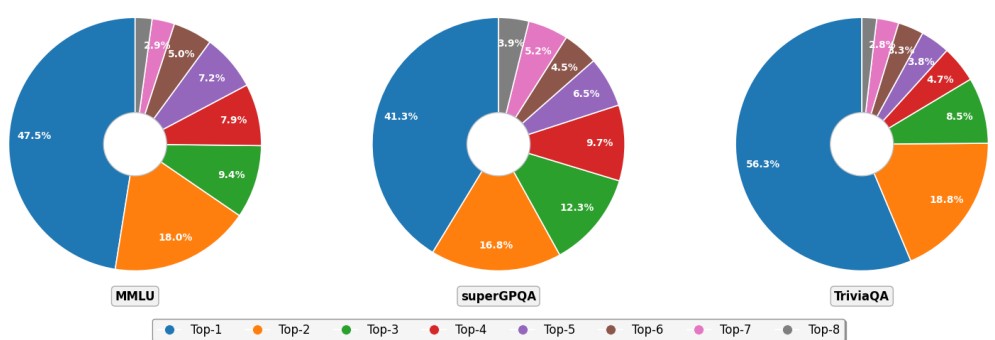

Figure 2: Distribution of minimum top-k retrieval requirements for correct responses using Llama3-8B-Instruct model on different datasets.

trieved across multiple user queries, and a small fraction of text chunks dominate the retrieval requests. As shown in Fig. 1a, we observe power-law distributions in text chunks popularity on the MMLU dataset (Hendrycks et al., 2020), where the most frequently accessed 10% of text chunks satisfy 80% of all questions under top-1 retrieval[1]. This skewed access pattern indicates substantial redundant computation during LLM inference, as the same contextual information is processed repeatedly for different user queries. The second inefficiency stems from over-allocation of context within individual queries, regardless of their complexity. Although LLMs consistently benefit from expanded contextual information, accuracy improvements follow a pattern of diminishing marginal utility with additional retrieved text chunks. We validate this intuition using Llama3-8B-Instruct (Dubey et al., 2024b) on MMLU, SuperGPQA (Du et al., 2025), and TriviaQA (Joshi et al., 2017) datasets, as shown in Fig. 2. By analyzing minimal knowledge requirements for accurate model predictions, we observe that over 60% of queries require only minimal context, whereas only approximately 3% need top-8 retrieval. This distribution highlights a critical inefficiency: static deep retrieval incurs unnecessary computational costs on simple queries while potentially degrading accuracy through contextual noise. These findings illuminate a fundamental optimization challenge in the RAG system: ***How can we achieve both computational efficiency and performance gains simultaneously?***

Caching represents a promising solution to address computational redundancy in RAG systems by reusing previously computed representations (*i.e.*, KV cache). Recent advances, including vLLM (Kwon et al., 2023), SGLang (Zheng et al., 2024), and RAGCache (Jin et al., 2024), employ prefix caching to store key-value representations of processed text chunks. While maintaining generation quality equivalent to full recomputation, these methods require exact sequence matching, leading to poor hit rates with longer contexts and positional variations. Independent chunk caching approaches attempt more flexible strategies. PromptCache (Gim et al., 2024) achieves higher efficiency through independent chunk caching but sacrifices accuracy by ignoring cross-chunk attention.

---

[1]For top-2 and top-3 retrieval, we treat each unique combination of retrieved text chunks as a distinct context unit for cumulative distribution analysis.

CacheBlend (Yao et al., 2025) partially restores cross-chunk attention via selective recomputation, yet applies uniform recomputation ratios across all chunks without considering the heterogeneous attention characteristics across different chunks. Furthermore, all prior work assumes static top-k retrieval, missing query-adaptive optimization opportunities that could enable simultaneous efficiency and accuracy improvements.

In this paper, we present an adaptive caching framework that addresses both computational redundancy and contextual over-provisioning in RAG systems through two complementary mechanisms. We first design a cache-aware partial recomputation method that profiles attention patterns to construct multiple cache variants per text chunk, selecting minimal recomputation strategies during reuse. Then, we introduce an adaptive context augmentation strategy that incrementally expands retrieval depth using lightweight confidence estimation to determine optimal context length for each user query. Evaluation across multiple models and datasets shows that we achieve 1.4x~5.0x TTFT reduction over state-of-the-art RAG caching systems while maintaining accuracy.

## 2 BACKGROUND AND RELATED WORK

Autoregressive Transformers execute inference in two distinct phases. In the prefill phase, the model processes the entire input sequence, performing self-attention across all tokens and materializing per-layer KV caches. In the subsequent decode phase, tokens are generated step by step while attending to this cached state. By reusing the stored projections of preceding tokens, the KV cache eliminates redundant recomputation of the prefix and enables efficient autoregressive generation.

RAG extends this pipeline by incorporating external evidence. A retriever encodes the user query, searches a corpus, and returns the top-$k$ passages (Ram et al., 2023; Siriwardhana et al., 2023; Jiang et al., 2023). The generator concatenates the query and retrieved passages, tokenizes the combined sequence, and applies the same prefill–decode process: prefill constructs KV entries for all tokens, and decode reuses them to produce the answer. However, concatenation markedly lengthens the prompt, increasing both attention cost and KV overhead in proportion to sequence length.

As a result, prefill dominates serving latency, raising TTFT and reducing throughput under load. Moreover, much of the additional computation is not essential for factual grounding, such as interactions among irrelevant passages or regions with low query attention. The fundamental bottleneck is thus the cost of full prefill and KV materialization over long contexts, motivating mechanisms that preserve only query–evidence interactions while avoiding redundant computation.

**General LLM Inference Systems.** vLLM (Kwon et al., 2023) accelerates generic serving via PagedAttention with block-wise KV paging and sharing; Orca (Yu et al., 2022) scales distributed decoding through iteration-level scheduling; prefill–decode disaggregation, as in DistServe (Zhong et al., 2024) and SplitWise (Patel et al., 2024), separates phases across GPUs to mitigate interference; and FlexGen (Sheng et al., 2023) expands effective capacity by aggregating memory and computation from the GPU, CPU, and disk. These approaches reduce phase contention and memory pressure but treat prompts as monolithic sequences, leaving them ill-suited to RAG's retrieval-induced redundancy and leading to suboptimal performance.

**Retrieval Optimization.** Sparse retrievers such as TF–IDF (Ramos et al., 2003) and BM25 (Robertson et al., 2009) enable efficient lexical matching, while dense retrievers leverage learned embeddings for higher recall at greater cost (Karpukhin et al., 2020). On top of these, rerankers refine first-stage results to improve precision with moderate overhead (Sun et al., 2023; Pradeep et al., 2023; Santhanam et al., 2021). These techniques focus on improving retrieval quality, whereas our method leaves the retrieved set unchanged and targets efficiency in post-retrieval processing.

**Context Reusing.** Caching mechanisms amortize the prefill cost by reusing KV states. Prefix caching, as in SGLang (Zheng et al., 2024), CachedAttention (Yao et al., 2025), and RAGCache (Jin et al., 2024), achieves fidelity but relies on exact prefix matches, resulting in low hit rates under long and variable RAG prompts. To alleviate this limitation, independent chunk caching relaxes matching: PromptCache (Gim et al., 2024) caches blocks independently but discards cross-chunk attention, thereby compromising accuracy, while CacheBlend (Yao et al., 2025) reintroduces interactions via selective recomputation yet applies uniform ratios oblivious to heterogeneous attention patterns. Our approach addresses these gaps by incorporating attention-aware cache variants with minimal recomputation to preserve fidelity, and by employing confidence-guided, per-query adaptive expan-

sion. This design reduces redundant computation, lowers long-context overhead, and significantly improves both TTFT and throughput.

## 3   ADACACHE

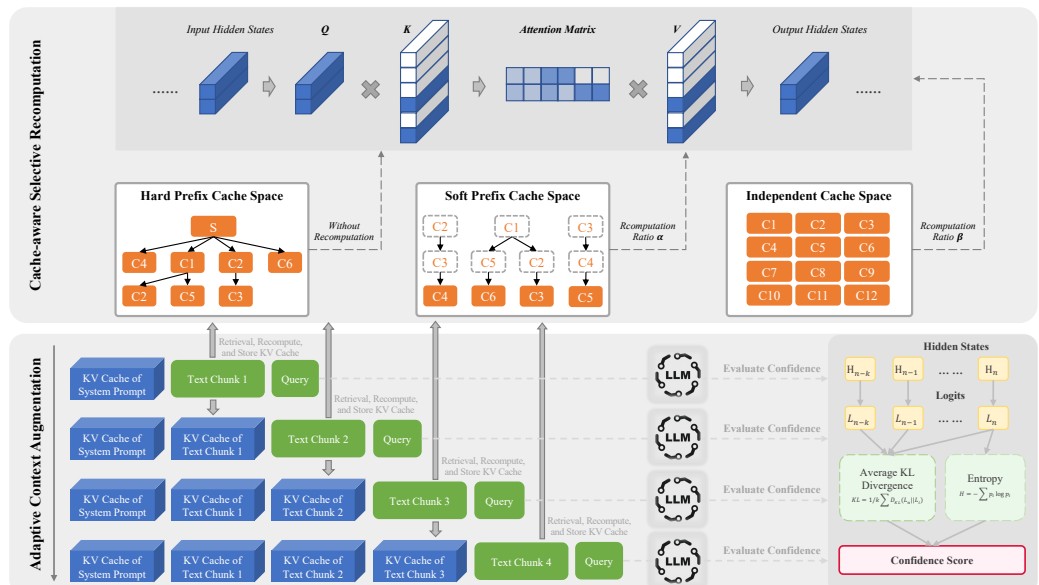

Figure 3: The Overview of AdaCache. It consists of two complementary modules. **Cache-aware selective recomputation** (upper) maintains three hierarchical cache spaces: (1) Hard prefix cache requires exact prefix matching and stores KV cache for all chunks along the matching path (solid boxes), enabling direct reuse without recomputation; (2) Soft prefix cache matches only effective prefix chunks requiring partial recomputation at ratio $\alpha$, where solid boxes represent cached entries while dashed boxes indicate prefix dependencies without storage; (3) Independent cache performs chunk-level matching with higher recomputation ratio $\beta$. The top attention diagram shows selective recomputation where 2 out of 6 tokens (blue solid blocks) are recomputed while the remaining 4 tokens reuse cached KV states. **Adaptive context augmentation** (lower) incrementally expands prompts by adding one text chunk at a time, evaluating confidence after each addition using a composite metric combining average KL divergence across the last few layers and output entropy, terminating when sufficient confidence is achieved or maximum context is reached.

### 3.1   CACHE-AWARE SELECTIVE RECOMPUTATION

**Attention Analysis.**   We begin by analyzing chunk-level attention patterns to understand inter-chunk dependencies in RAG contexts. The augmented prompt is segmented into discrete chunks: *[system prompt, text chunk 1, ..., text chunk k, query]*, and we aggregate attention weights of each layer into chunk granularity. Fig. 4 demonstrates two distinct attention distributions across model depth during Qwen3-8B model inference [2]. Early layers (1-18) show localized patterns where each chunk primarily attends to its predecessor, while deeper layers (19-36) exhibit attention sink phenomena, with certain chunks capturing most attention from subsequent chunks. This pattern reveals that only a subset of chunks serves as effective prefixes, enabling joint caching of partial prefix sequences to restore cross-chunk dependencies lost in independent chunk caching.

**Hierarchical Cache.**   Based on the observed attention patterns, we establish a three-tier cache hierarchy that systematically balances cache utilization efficiency against generation quality. *Hard*

---

[2] We validated these chunk-level attention patterns across Llama3-8B-Instruct, Qwen3-4B, and Qwen3-8B models on MMLU, TriviaQA, and SuperGPQA datasets, observing consistent behaviors, though the specific chunk positions serving as attention sinks vary across different contexts.

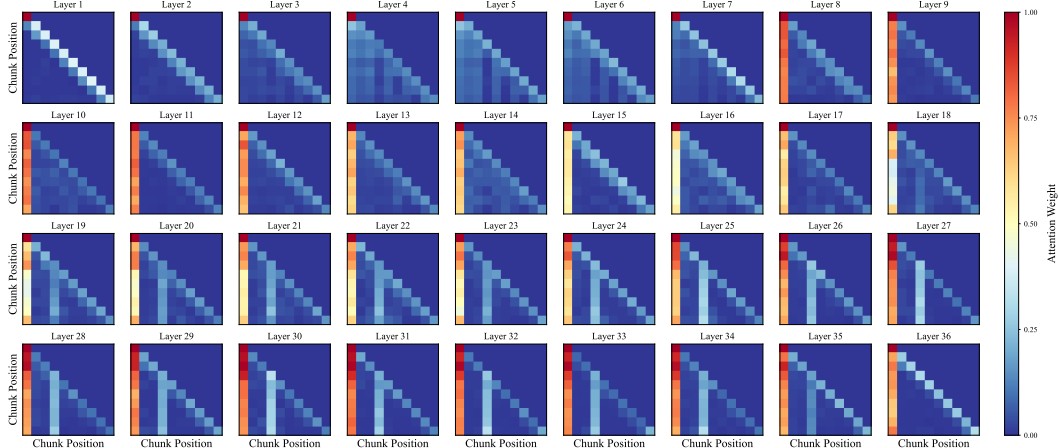

Figure 4: Chunk-level attention patterns during Qwen3-8B model inference with Top-8 retrieval. The first and last columns in each subplot correspond to the system prompt and user query, respectively. Layers 1-18 exhibit localized attention where chunks predominantly focus on their immediate predecessors with sparse attention to distant chunks. In layers 19-36, an attention sink phenomenon emerges where the 3rd chunk captures the majority of attention from subsequent chunks.

*Prefix Cache* requires exact prefix sequence matching, making it the most restrictive but accuracy-preserving tier. Due to the causal attention mask in autoregressive inference, exact prefix matches guarantee computational equivalence to full recomputation, thereby preserving perfect generation quality when cache hits occur. However, this strict requirement significantly constrains cache utilization. *Soft Prefix Cache* relaxes the matching constraint to effective prefix matching, where only the sink chunk or predecessor chunk needs to match for cache reuse. This design leverages our attention analysis findings: since attention primarily flows from these key chunks, partial prefix matching can maintain most cross-chunk dependencies. *Independent Cache* provides the fallback mechanism when prefix matching fails entirely. Individual text chunks are precomputed and stored independently without prefix dependencies. It maximizes cache hit rates but poses the greatest risk for accuracy preservation, as cross-chunk attention dependencies must be reconstructed during LLM inference.

**Cache Reusing and Recomputation.** Building on previous work (Yao et al., 2025), only a subset of tokens within each chunk exhibit significant cross-chunk attention, leading to substantial KV states deviations compared to those in the independent cache. Critically, this sparsity pattern exhibits layer-wise consistency: tokens with the highest KV deviations in one layer are likely to have the highest deviations in subsequent layers. This insight enables efficient selective recomputation by identifying attention-critical tokens through first-layer analysis and applying the same selection across all layers. We determine recomputation candidates by analyzing cross-chunk attention ratios in the initial layer, selecting tokens with the highest proportion of cross-chunk attention weights.

Rather than a uniform recomputation across all chunks in context, we adapt the recomputation ratio based on available cache matches. The KV states of each chunk may have multiple cached variants stored under different prefix contexts. We retrieve from cache spaces in a hierarchical order with progressively relaxed matching constraints.

We first query the hard prefix cache for exact matches, where any chunks along the matched prefix path can be directly reused without recomputation. When exact matching fails, we examine the soft prefix cache for effective prefix alignment. Note that soft prefix chunks serve only as cache key identifiers without maintaining separate cache entries. Successful soft matching requires recomputing $\alpha$ fraction of tokens to restore global cross-chunk attention[3]. If no cached KV states exist in

---

[3]In the first half of model layers, effective prefixes correspond to predecessor chunks, while in the second half, they correspond to both sink chunks and predecessor chunks. We identify sink chunk positions by analyzing attention matrices at transition layers: chunks before the sink chunk require predecessor-based matching, while chunks after the sink chunk use both sink chunks and predecessor chunks as their effective

---

**Algorithm 1** Adaptive Context Augmentation

---

**Require:** System prompt $s$, query $q$, retrieved text chunks $[c_1, \ldots, c_k]$, confidence threshold $\tau$

1: **for** $i = 1$ to $k$ **do**
2:     $p \leftarrow s + [c_1, \ldots, c_i] + q$                         $\triangleright$ Construct context-augmented prompt
3:     **for** $j = 1$ to $i - 1$ **do**
4:         Retrieve hash($[c_1, \ldots, c_j]$) in *Hard Prefix Cache*    $\triangleright$ Reuse KV states w/o recomputation
5:     **end for**
6:     **if** hash(effective_prefix($[c_1, \ldots, c_i]$)) $\in$ *Soft Prefix Cache* **then**
7:         $\rho \leftarrow \alpha$                            $\triangleright$ Lower recomputation ratio for *Soft Prefix Cache*
8:     **else**
9:         $\rho \leftarrow \beta$                            $\triangleright$ Higher recomputation ratio for *Independent Cache*
10:    **end if**
11:    $T \leftarrow \rho$ fraction of tokens in $c_i$    $\triangleright$ Selected tokens with the highest cross-chunk attention
12:    $O \leftarrow \text{Prefill}(T)$                               $\triangleright$ Generate logits for last $l$ layers
13:    $\text{conf} \leftarrow \lambda \cdot \widehat{KL}(O_{1..l-1}, O_l) + (1 - \lambda) \cdot \widehat{H}(O_l)$      $\triangleright$ Evaluate the confidence
14:    **if** $\text{conf} > \tau$ **then**
15:        Execute decoding                      $\triangleright$ Stop context augmentation
16:        **break**
17:    **end if**
18:    Update KV states of $c_i$ with hash($[c_1, \ldots, c_i]$) in *Hard Prefix Cache*
19: **end for**

---

the soft prefix cache space, we turn to the independent cache with recomputation ratio $\beta$ ($\beta > \alpha$) to reconstruct discarded cross-chunk dependencies. This cache reuse approach achieves an optimal efficiency-accuracy trade-off through adaptive token recomputation that responds to varying prefix match conditions: exact, partial, or absent.

## 3.2 ADAPTIVE CONTEXT AUGMENTATION

Algorithm 1 presents the process of adaptive context augmentation (ACA) with cache-aware recomputation. Rather than concatenating all top-k retrieved text chunks into the user prompt simultaneously, we employ an incremental augmentation strategy that progressively incorporates one chunk at a time until reaching the k-th chunk or achieving sufficient confidence. While this approach necessitates multiple forward passes for the same query, it eliminates redundant context computation through strategic caching. At each iteration, we only recompute the KV states for the newly added chunk, storing them in the hard prefix cache space for reuse in subsequent context augmentation. This ensures that all previously processed chunks maintain cache hits, dramatically reducing computational overhead. Importantly, ACA does not introduce any additional retrieval overhead. The top-$k$ retrieval step is executed once per query, and the augmentation loop then operates solely within the prefill phase using the already-retrieved text chunks.

To decide whether augmentation should terminate, we employ a composite confidence metric combining two complementary uncertainty measures. First, we compute the average KL divergence between the logits of the last $l$ layers and the final layer, capturing internal reasoning consistency. If the model can accurately infer the answer from the current context, its logit distribution should converge early across layers. Second, we calculate the entropy of the final token distribution, reflecting output uncertainty. We normalize both the average KL divergence and entropy to $[0, 1]$, then compute a weighted confidence score, with weights determined through optimization on the validation set. This dual-metric balances stability with predictive certainty, providing a more robust confidence estimate. Notably, the confidence metric is computationally lightweight. In practice, AdaCache computes logits only for the last 4 layers and for the final token rather than the full context, which keeps the overhead negligible at less than $1\%$ of the prefill cost.

ACA reduces computational and memory demands by avoiding excessive context allocation for simple queries. Given $k$ retrieved text chunks of length $l_c$ tokens each, a query of length $l_q$ tokens,

---

prefixes. Including predecessor chunks prevents cumulative errors that would arise from inconsistency with the predecessor-based matching used in the first half of layers.

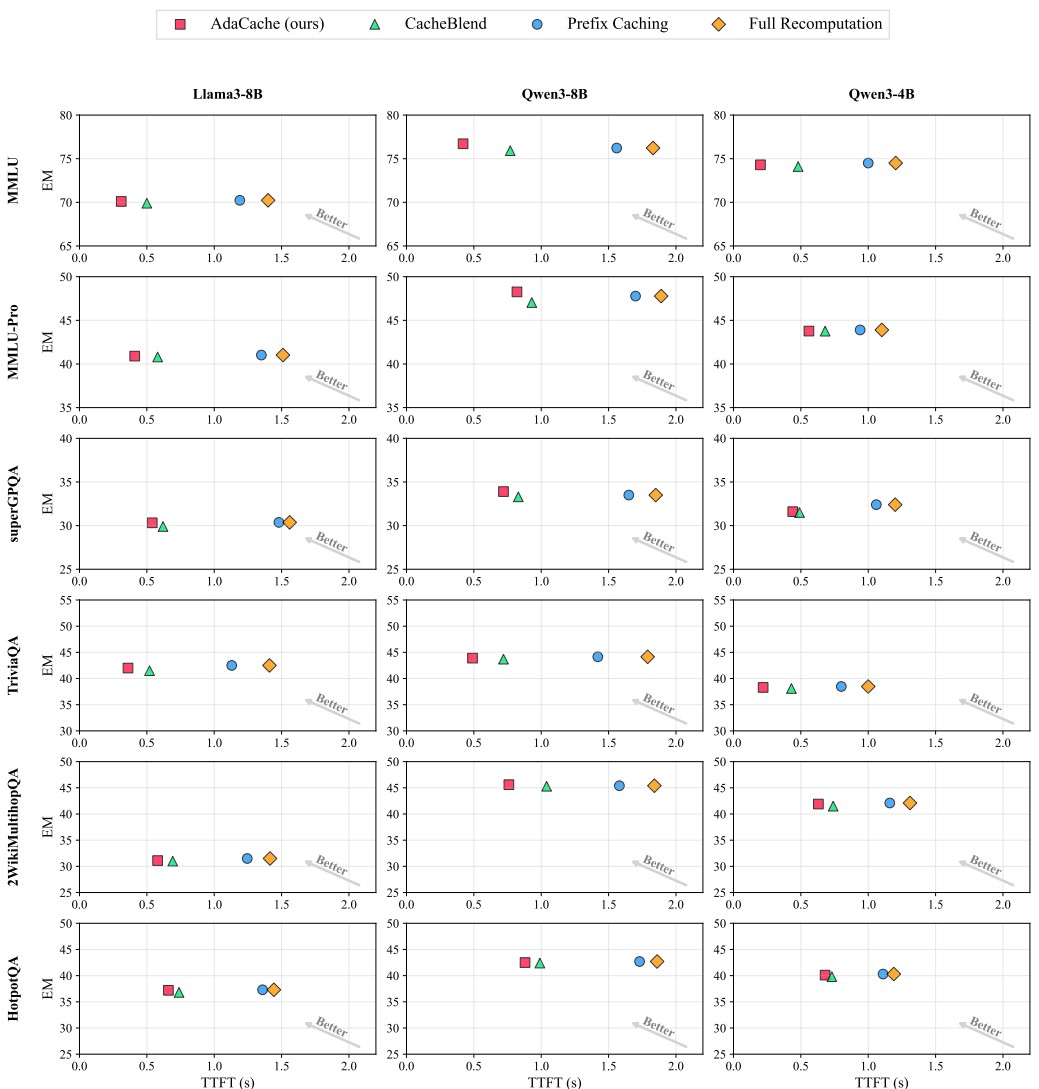

Figure 5: Comparison of Time-to-Tirst-Token (TTFT) and generation quality between AdaCache and baseline methods across six datasets and three models.

and early termination at step $t$, ACA processes at most $t \cdot (l_c + l_q)$ tokens. It yields substantial savings in computation and memory compared to static context augmentation, which requires processing $k \cdot l_c + l_q$ tokens [4].

## 4 EVALUATION

### 4.1 EVALUATION METHODOLOGY

**Models and Hardware Settings.** We evaluate AdaCache using *Llama-3-8B-Instruct* (Dubey et al., 2024a), *Qwen3-4B*, and *Qwen3-8B* (Yang et al., 2025) models. Experiments are conducted

---

[4]Retrieved text chunks are typically longer than queries, with 512 tokens being a common chunk size while queries usually contain fewer than 128 tokens. For example, with top-6 retrieval, static context augmentation processes 3,200 tokens per query during the prefill phase. In contrast, given the empirical expectation of $t = 2$ text chunks required across the dataset, ACA processes only 1,280 tokens, achieving 2.5x reduction in computational cost.

on a server equipped with 128 CPU cores (2×Intel Xeon Gold 6530), 512 GB of host memory, and NVIDIA RTX 6000 Ada GPU with 48 GB memory. Data transfers between the CPU and GPU are carried out over the PCIe 4.0×16 interface.

**Corpus and Datasets.** We use the Wikipedia dataset[5] as our knowledge base. Prior to embedding, all documents are segmented into chunks of size 512 tokens. Each chunk is then encoded using the *e5-base-v2* embedding model. For vector search, we leverage the FAISS library to construct an inverted file (IVF) index with 1024 clusters, and set the default top-k retrieval to 6. AdaCache is evaluated on several rigorous benchmark datasets, including MMLU (Hendrycks et al., 2020), MMLU-Pro (Wang et al., 2024), SuperGPQA (Du et al., 2025), TriviaQA (Joshi et al., 2017), 2Wiki-MultihopQA (Ho et al., 2020), and HotpotQA (Yang et al., 2018), which span general knowledge, open-domain reading comprehension, and advanced reasoning.

**Baselines.** We compare AdaCache with three baselines: *(i) Full Recomputation*, where the raw text is fed into the LLM and the KV cache for all tokens is computed during prefill; *(ii) Prefix Cache* (Jin et al., 2024), which leverages SGLang (Zheng et al., 2024) to identify frequently used prefix chunks and persist their KV caches in RAM and SSD, while non-prefix tokens are still computed during prefill. For fairness, we optimistically assume no delay when loading from RAM/SSD to GPU, which favors this scheme relative to real deployments; and *(iii) Selective Recomputation*, which adopts CacheBlend (Yao et al., 2025) to reuse precomputed KV caches of all chunks, while selectively recomputing in each layer a small subset of high-deviation tokens to restore cross-chunk attention.

**Metrics.** We evaluate models on both accuracy and responsiveness. Accuracy is measured by *Exact Match (EM)*, the fraction of predictions that exactly match a normalized reference answer. Responsiveness is measured by *Time-To-First-Token (TTFT)*, the wall-clock latency from request submission to the emission of the first output token. We report results across repeated runs under controlled hardware and inference settings.

## 4.2 EXPERIMENTAL RESULTS

Naive RAG systems recompute KV caches for every new request and its retrieved context. Ada-Cache achieves substantial TTFT reductions of 3.12× on average and up to 6.02× compared to full recomputation while preserving nearly identical generation quality. The performance gains derive from AdaCache's dual optimization strategy, which simultaneously eliminates cross-request computational redundancy in overlapping contexts while preventing unnecessary context augmentation for simple queries. Notably, AdaCache occasionally surpasses full recomputation in prediction accuracy, as excessive contextual information can introduce noise that degrades model reasoning. Guided by model output confidence, AdaCache ensures that the minimal sufficient context contributes to the generation process.

AdaCache demonstrates 2.69× average and up to 5.0× performance improvements over prefix caching. While prefix caching eliminates redundant computation of overlapping prefixes and maintains identical generation quality to full recomputation, exact prefix matching limits its effectiveness with longer contexts or dynamic positioning of retrieved chunks. AdaCache addresses these limitations with a hierarchical cache architecture (*i.e.*, hard prefix cache, soft prefix cache, and independent caches), enabling more flexible cache reuse.

CacheBlend leverages independent caching to achieve substantial improvements in cache hit rates, employing selective recomputation to maintain cross-chunk attention and preserve generation quality. In comparison, AdaCache delivers 1.32× on average and up to 2.34× TTFT improvements over CacheBlend with marginally superior generation quality. AdaCache analyzes inter-chunk attention patterns across layers and constructs soft prefix caches, enabling flexible hierarchical caching that reduces token-level recomputation and decreases TTFT. Additionally, adaptive context selection reduces computational waste from non-contributory text chunks.

---

[5]We use the `wikimedia/wikipedia` dataset on Hugging Face, which contains cleaned articles from the official Wikipedia dumps. Each subset corresponds to one language and consists of a single training split with markdown and references removed. In our experiments, we adopt the English subset released on 2023-11-01.

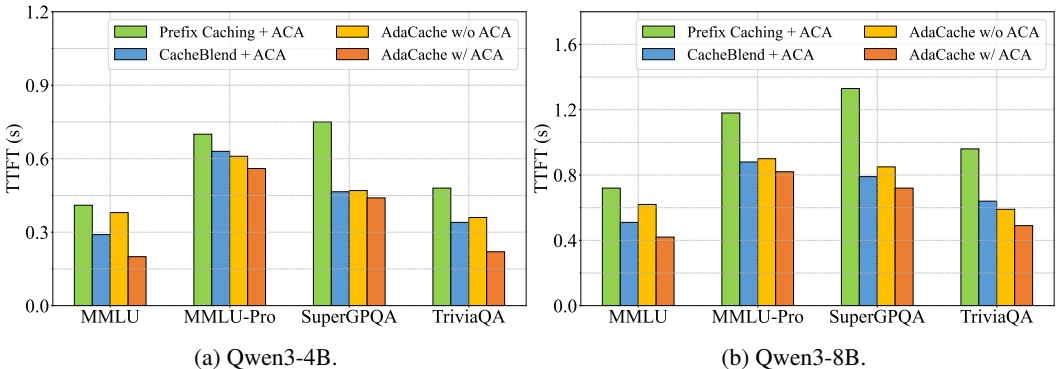

(a) Qwen3-4B.  (b) Qwen3-8B.

Figure 6: TTFT comparison of caching strategies combined with and without ACA across different datasets and models.

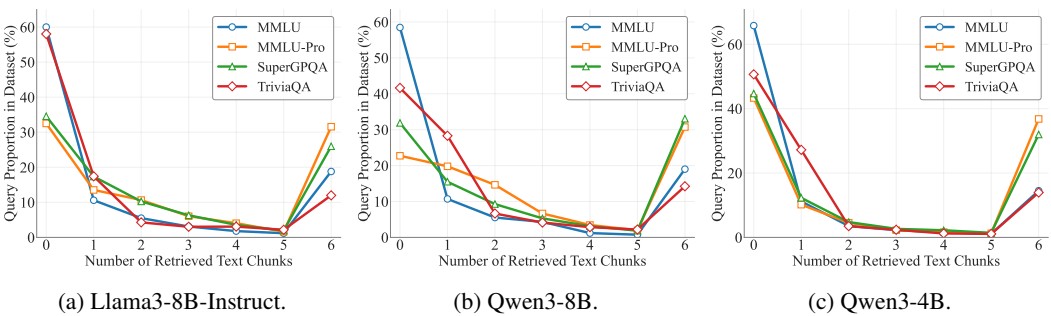

(a) Llama3-8B-Instruct.  (b) Qwen3-8B.  (c) Qwen3-4B.

Figure 7: The context length distribution determined by adaptive context augmentation across different datasets and models.

**Ablation Study.** To isolate the performance contributions of cache-aware selective recomputation and adaptive context augmentation (ACA), we compare four configurations: Prefix Caching combined with ACA, CacheBlend combined with ACA, AdaCache without ACA, and the full AdaCache system (Fig. 6). ACA integrates smoothly with different caching mechanisms, and when applied to Prefix Caching or CacheBlend, it consistently reduces TTFT across datasets and models, delivering average speedups of 1.65× and 1.22×, respectively.

The full AdaCache system retains substantial performance advantages over ACA-enhanced baselines, achieving average speedups of 1.76× over Prefix Caching and 1.23× over CacheBlend. Notably, cache-aware selective recomputation alone occasionally outperforms even ACA-enhanced baselines. These results demonstrate the effectiveness of our hierarchical caching design and its tight integration with ACA.

**Context Length Distribution.** To better understand the performance improvements of Adaptive Context Augmentation (ACA), we analyze the distribution of context lengths identified by ACA during model inference. As shown in Fig. 7, a consistent pattern emerges across all three models and four datasets: the majority of queries require minimal context augmentation, while queries requiring longer contexts become increasingly rare. The sharp spike at maximum length includes queries that remain unanswerable even when provided with the complete top-6 retrieved text chunks, indicating persistently low confidence throughout the ACA process.

The performance gains from ACA correlate strongly with this distribution pattern. Datasets exhibiting more pronounced head-heavy distributions with smaller tail proportions yield greater improvements. MMLU and TriviaQA demonstrate more skewed distributions compared to MMLU-Pro and SuperGPQA, with correspondingly higher relative performance gains. Specifically, AdaCache achieves 1.95× and 1.62× average TTFT reduction over CacheBlend on MMLU and TriviaQA,

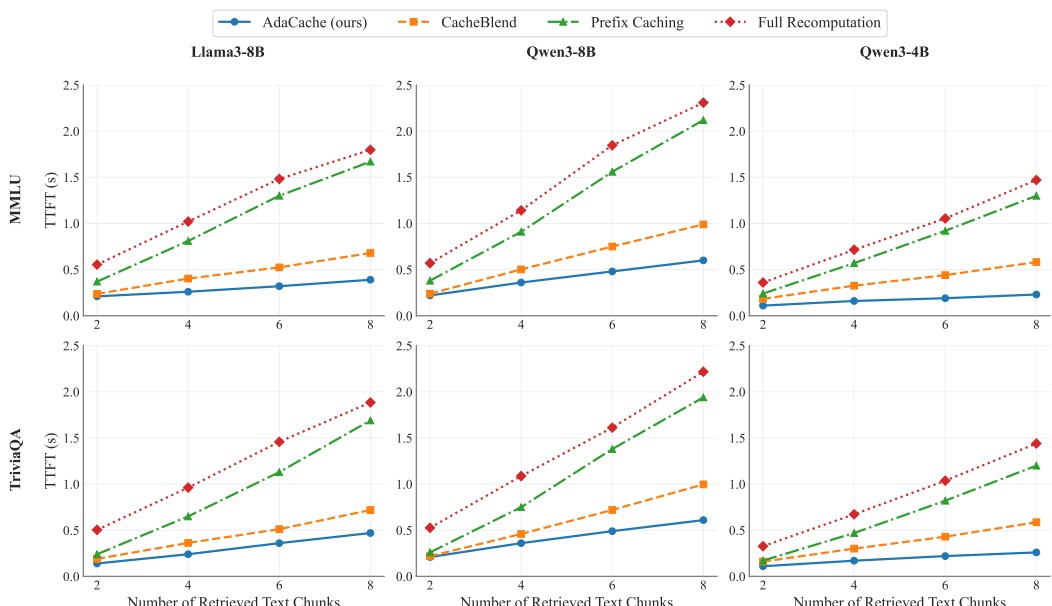

Figure 8: Comparison of TTFT between AdaCache and baseline methods across different top-k retrieval.

respectively, across three models, compared to more modest improvements of 1.25× and 1.14× on MMLU-Pro and SuperGPQA respectively.

**Performance Across top-k Retrieval Settings.** Fig. 8 demonstrates the performance comparison between AdaCache, CacheBlend, Prefix Caching, and Full Recomputation across varying top-k retrieval configurations[6]. At top-2 retrieval, the performance gap between Prefix Caching, CacheBlend, and AdaCache remains modest. Prefix Caching achieves a substantial TTFT reduction compared to Full Recomputation due to relatively high cache hit rates in short context scenarios.

However, as context expands, a clear performance divergence emerges. Prefix Caching suffers dramatic degradation, with TTFT improvements declining from an average of 1.76× at top-2 to merely 1.13× at top-8 retrieval, reflecting the fundamental limitation of strict prefix matching in long context scenarios. In contrast, AdaCache exhibits superior context scalability, with performance gains improving from an average of 2.93× to 4.67× over full recomputation. While CacheBlend's independent caching strategy substantially improves cache hit rates for long contexts compared to Prefix Caching, AdaCache achieves fundamentally better context scalability by combining hierarchical caching with adaptive context augmentation.

## 5 CONCLUSION

We present AdaCache, a comprehensive framework that addresses fundamental computational inefficiencies in RAG systems through dual optimization strategies: cache-aware partial recomputation that profiles attention patterns to construct selective cache variants, and adaptive context augmentation that dynamically determines optimal retrieval depth via lightweight confidence estimation. Our approach tackles two key inefficiencies observed in current RAG systems: the power-law distribution of context reuse across queries, where 10% of chunks satisfy 80% of retrieval requests, and the over-allocation of context, where 60% of queries require only minimal retrieval. Comprehensive evaluation demonstrates that AdaCache achieves 1.4×∼5.0× TTFT reduction over state-of-the-art RAG caching systems while maintaining generation quality. Notably, our adaptive context augmentation enables seamless integration with existing caching strategies while exhibiting superior context scalability.

---

[6]For AdaCache, top-k refers to the maximum available context length during adaptive context augmentation.

ACKNOWLEDGMENTS

This research is supported by the NTU startup grant and the RIE2025 Industry Alignment Fund –Industry Collaboration Projects (IAF-ICP) (Award I2301E0026), administered by A*STAR, as well as supported by Alibaba Group and NTU Singapore through Alibaba-NTU Global e-Sustainability CorpLab (ANGEL).This research is also supported by the Ministry of Education, Singapore, under its Academic Research Fund Tier 2 (Award MOE-T2EP20125-0005).

ETHICS STATEMENT

Our work adheres to the ICLR Code of Ethics. Our research does not involve human subjects, sensitive personal data, or any identifiable information. All datasets used in our experiments are publicly available and widely adopted in prior literature. We strictly followed the terms of use of these datasets and ensured that no proprietary or private information was accessed or disclosed. The methods developed are intended purely for academic research and are not designed to produce harmful applications. We are committed to promoting fairness, transparency, and reproducibility in machine learning research, and we release our results in compliance with community standards of research integrity.

REPRODUCIBILITY STATEMENT

We provide full details to support reproducibility. The AdaCache framework, including cache-aware recomputation and adaptive context augmentation, is specified in Section 3 with pseudocode and design assumptions. Experimental settings, datasets, preprocessing, evaluation metrics, and baseline configurations are described in Section 4. Model architectures, and hardware settings are reported to allow replication of latency and throughput measurements.

THE USE OF LARGE LANGUAGE MODELS (LLMS)

We used large language models as the general-purpose assistive tool during the preparation of this paper. Its contributions were limited to improving grammar, polishing wording, and suggesting alternative phrasings for clarity and conciseness. The research ideas, methodological design, experimental implementation, analysis, and final interpretations were entirely conceived and executed by the authors.

LLMs were not used for generating novel research content, fabricating facts, or conducting scientific reasoning. All technical descriptions, results, and conclusions presented in the paper are the sole responsibility of the authors.

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
