# OpenReview forum: "AdaCache: Adaptive Caching and Context Augmentation for Efficient LLM Serving"
_ICLR.cc/2026/Conference — ICLR 2026 Poster_

### Official Review · Reviewer_8B2T · 2025-10-31

**Soundness:** 2
**Presentation:** 2
**Contribution:** 2
**Rating:** 4
**Confidence:** 5

**Summary:**

This paper points two major inefficiencies in current RAG systems: (1) cross-query context overlap, where identical text chunks from the external knowledge base are repeatedly retrieved across multiple user queries, and (2) a small fraction of text chunks dominate the retrieval requests. Based on these two findings, this paper aims to address both computational inefficiency simultaneously. AdaCache is then proposed to speed up the loading of contextual information. Experimental results on Llama 3 8B, Qwen 3 8B, and Qwen 3 4B show the proposed method is faster than CacheBlend and Prefix Caching in terms of TTFT with close correctness.

**Strengths:**

* This work is well-motivated and easy to follow.

* Experiments support the effectiveness of the proposed AdaCache methodology.

**Weaknesses:**

* The current experiments are based on MMLU, MMLU-Pro, superGPQA, and TriviaQA, where the question can be answered by single factual chunk. Evaluations on more challenging RAG scenarios that require handling long and/or complex contextual information could be added for revealing the limitation of AdaCache.

* This framework requires the manipulation of KV-cache and attention values, limiting its applications to open-weight models.

* Comparison with [TurboRAG](https://arxiv.org/abs/2410.07590) could be added.

**Questions:**

* Can these method be seamlessly integrated with dense retrieval model, where the embedding of each chunk is also the cache?

* The quality of Figure 3 could be further improved for readability.

---

> ### Author Response · Authors · 2025-11-22
>
> **We sincerely thank the reviewer for the constructive feedback. We appreciate the recognition of our contributions and would like to address the concerns raised:**
>
>  > ### **W1: Evaluations on more challenging RAG scenarios.**
>
> We conducted experiments on two popular multi-hop QA datasets using Qwen3-4B and Qwen3-8B. These datasets require synthesizing information across multiple retrieved documents to answer complex queries. Results demonstrate that AdaCache consistently outperforms Full Recomputation, Prefix Cache, and CacheBlend across all datasets and models while maintaining comparable generation quality.
>
> | Dataset | Model     | Method          | EM | TTFT (s) |
> | ----------- |-----------|-----------------|--------|----------|
> |  **2WikiMultihopQA** | **Qwen3-4B** | Full Recompute   | 42.1      |  1.31   |
> | |           | Prefix Caching    | 42.1      |   1.16   |
> | |           | CacheBlend        | 41.5      |  0.74   |
> | |           | **AdaCache (Ours)** | 41.9    | **0.63** |
> | | **Qwen3-8B** | Full Recompute   | 45.4      |  1.84   |
> | |           | Prefix Caching    | 45.4      |  1.58  |
> | |           | CacheBlend        | 45.3      |  1.04   |
> | |           | **AdaCache (Ours)** | 45.6    | **0.76** |
> | **HotpotQA** | **Qwen3-4B** | Full Recompute   | 40.3      | 1.19    |
> | |           | Prefix Caching    | 40.3      | 1.11     |
> | |           | CacheBlend        | 39.8      | 0.73     |
> | |           | **AdaCache (Ours)** | 40.1    | **0.68** |
> | | **Qwen3-8B** | Full Recompute   | 42.7      | 1.86     |
> | |           | Prefix Caching    | 42.7      | 1.73     |
> | |           | CacheBlend        | 42.4      | 0.99    |
> | |           | **AdaCache (Ours)** | 42.5    | **0.88** |
>
>
> > ### **W2: Applicability beyond open-weight models.**
>
> We respectfully clarify that while AdaCache requires KV-cache manipulation, this does not limit its utility to open-weight models exclusively. Popular inference frameworks such as vLLM, SGLang, and LMCache all involve KV-cache management. AdaCache is a system-level optimization that LLM vendors can incorporate without exposing model weights. Closed-source providers can equally deploy AdaCache in their inference infrastructure to reduce serving costs and enhance service quality, just as they currently implement various KV-cache optimizations for efficiency.
>
>
>
> > ### **W3: Comparison with TurboRAG.**
>
> TurboRAG independently pre-computes KV caches for text chunks and reuses them during online serving. However, it breaks cross-chunk attention dependencies, necessitating model retraining on specially constructed datasets to adapt to sparse attention masks. This retraining process is both computationally expensive and time-consuming (e.g., the paper reports using 32 A100 GPUs). Furthermore, each new model architecture requires separate retraining, and without publicly released trained models, reproduction is challenging.
>
> In contrast, AdaCache is **training-free** and introduces only minimal overhead from confidence computation and profiling. Importantly, our adaptive context augmentation module is orthogonal to TurboRAG's caching strategy, combining both could further reduce KV-cache loading cost.
>
>
> > ### **Q1: Integration with dense retrieval models.**
>
> Our evaluation employs dense retrieval for context augmentation. AdaCache's design is retriever-agnostic and can seamlessly integrate with any dense or sparse retrieval pipeline. In our current implementation, we don't cache chunk embeddings because retrieval latency is significantly smaller compared to LLM inference costs. However, caching embeddings would further reduce end-to-end latency when deploying embedding models.
>
> > ### **Q2: Figure 3 could be further improved for readability.**
>
> Thank you for the feedback. We apologize for the readability issues and will improve Figure 3 for clarity in the revised manuscript.

---

> ### Author Response · Authors · 2025-12-03
> **Additional experiments and clarifications**
>
> > ### **W1: Evaluations on more challenging RAG scenarios.**
>
> We further conduct experiments using the Llama-3-8B-Instruct model on the 2WikiMultihopQA and HotpotQA datasets. Compared with Full Recompute, Prefix Caching, and CacheBlend, AdaCache achieves average speedups of 2.16×, 1.95×, and 1.18×, respectively, while maintaining comparable generation quality.
>
> | Model |   Dataset  | Method          | EM | TTFT (s) |
> | ----------- |-----------|-----------------|--------|----------|
> | **Llama-3-8B-Instruct**| **2WikiMultihopQA** | Full Recompute   | 31.5  |  1.41|
> | |           | Prefix Caching    |   31.5  |   1.25   |
> | |           | CacheBlend        |   31.0    | 0.69   |
> | |           | **AdaCache (Ours)** |  31.1   | **0.58** |
> | | **HotpotQA** | Full Recompute   |   37.3    |  1.44   |
> | |           | Prefix Caching    |    37.3   |   1.36   |
> | |           | CacheBlend        |   36.8    |   0.74  |
> | |           | **AdaCache (Ours)** |   37.2  | **0.66** |
>
> We have added the evaluations on these two multi-hop datasets in Section 4.2 (Figure 5) of the revised manuscript.
>
>
>
> > ### **Q2: Figure 3 could be further improved for readability.**
>
> We have improved Figure 3 in the revised manuscript with the following key modifications:
> 1. Enhanced the illustration of the soft prefix cache space by using dashed boxes to represent effective prefix dependencies, indicating that no actual cache entries are stored.
> 2. Added LLM icons to clearly indicate model inference at each iteration of the adaptive context augmentation process.
> 3. Annotated the newly added text chunk in each iteration with "Retrieval, Recompute, and Store KV Cache" to clarify the caching workflow.

---

### Official Review · Reviewer_DScb · 2025-10-31

**Soundness:** 3
**Presentation:** 3
**Contribution:** 3
**Rating:** 6
**Confidence:** 3

**Summary:**

This paper provides a efficient LLM Serving strage named AdaCache, which addresses two key inefficiencies of current RAG systems—redundant processing of frequent text chunks and uniform deep retrieval—via two optimization strategies. It uses cache-aware partial recomputation, which profiles attention patterns to build selective cache variants for flexible reuse while preserving cross-chunk dependencies. It also adopts adaptive context augmentation, dynamically determining optimal retrieval depth through lightweight confidence estimation to avoid unnecessary overhead on simple queries. Experiments across datasets and LLMs show AdaCache cuts Time-To-First-Token (TTFT) by 1.4x–5.0x versus state-of-the-art RAG caching systems, while maintaining generation quality.

**Strengths:**

Easy architecture and good  performance.

- Efficient Resource Utilization: By reusing cached KV states of frequent text chunks and avoiding over-retrieval for simple queries, AdaCache drastically reduces redundant computation. This translates to a 1.4x–5.0x TTFT reduction, greatly improving LLM serving throughput without wasting computational resources.​
- Quality Preservation: Its cache-aware partial recomputation preserves cross-chunk dependencies via attention pattern analysis and selective recomputation, while adaptive context augmentation uses a composite confidence metric to ensure sufficient context for accurate generation. Thus, it maintains generation quality even with efficiency gains.​
- Strong Adaptability: The three-tier cache hierarchy (hard prefix, soft prefix, independent cache) adapts to different prefix match conditions, and adaptive context augmentation adjusts retrieval depth based on query complexity. It works across diverse datasets and LLMs, showing broad applicability.

**Weaknesses:**

Some points can be optimized.

- Dependence on Attention Pattern Consistency.
The cache-aware recomputation relies on consistent layer-wise attention patterns (e.g., early localized attention, deep attention sinks). If LLMs have irregular attention distributions for certain tasks/domains, the cache variant design may fail, reducing reuse efficiency or accuracy.​
- Increasing System Complexity.
The three-tier cache management, selective recomputation ratio tuning, and composite confidence metric (KL divergence + entropy) add complexity to system implementation and maintenance. This raises the barrier for integration into existing RAG pipelines, especially for teams with limited engineering resources.​
- Potential Overhead in Incremental Augmentation.
Adaptive context augmentation requires multiple forward passes (adding one chunk at a time) for queries needing deep retrieval. Though cached reuse mitigates this, it may still introduce more latency than static retrieval for complex queries requiring full top-k chunks, offsetting efficiency gains.​
- Reliance on Confidence Metric Tuning.
 The confidence metric (weighted KL divergence and entropy) needs optimization on validation sets. Poor tuning (e.g., misaligned weights) could lead to premature termination (insufficient context, lower accuracy) or unnecessary augmentation (wasted resources), making performance unstable across unseen data.

**Questions:**

Large models preformance can be provided for 14b/30b/72b and so on.

---

> ### Author Response · Authors · 2025-11-22
>
> **We sincerely thank the reviewer for the constructive feedback. We appreciate the recognition of our contributions and would like to address the concerns raised:**
>
> > ### **W1: Dependence on Attention Pattern Consistency.**
>
> We validated chunk-level attention patterns illustrated in Figure 4 across multiple models (Llama3-8B-Instruct, Qwen3-4B, and Qwen3-8B ) and datasets (MMLU, MMLU-Pro, TriviaQA, SuperGPQA), observing consistent behaviors, though the specific chunk positions serving as attention sinks vary across different contexts.
>
> AdaCache’s caching mechanism builds on two key insights. First, within each text chunk, only a subset of tokens exhibit significant cross-chunk attention dependencies. These tokens require recomputation to correctly restore their KV representations, whereas the majority of tokens can safely reuse the precomputed KV cache. Second, tokens needing cross-chunk attention do not require full attention to the entire prefix context. Instead, they focus on a limited number of prefix chunks. By identifying and matching only these effective prefix chunks, AdaCache further reduces the recomputation ratio.
>
> Importantly, the attention sparsity is pervasive across different LLM architectures and tasks. Although the exact attention distribution may shift, the methodology of AdaCache—profiling prefix dependencies and capturing effective prefix chunks—remains robust.
>
>
> > ### **W2: Increasing System Complexity.**
>
> We implemented the system prototype of AdaCache on top of vLLM. AdaCache is modular where both cache-aware recomputation and adaptive context augmentation can function independently. AdaCache is also retriever-agnostic and integrates seamlessly with existing RAG pipelines.
>
> The hierarchical caching module extends prefix caching with multi-granularity cache keys to support flexible prefix matching. The composite confidence metric (weighted KL divergence + entropy) is lightweight, incurring less than 1% prefill overhead. It computes logits only for the last 3-4 layers on the final token. Additionally, the recomputation ratio is exposed as a tunable parameter, allowing users to balance generation quality and latency based on application-specific requirements.
>
>
> > ### **W3: Potential Overhead in Incremental Augmentation.**
>
> We clarify that adaptive context augmentation does not increase retrieval overhead. The top-k retrieval is executed once for each query, identical to all baseline methods. AdaCache then incrementally incorporates the already-retrieved chunks into the prompt during the prefill phase, not by repeated retrieval calls.
>
> For example, in our evaluation where all baselines use top-6 retrieval, AdaCache follows the same top-6 retrieval step. During the prefill phase, AdaCache begins with an empty context and progressively adds chunks (0→1→2→...→6) based on confidence-guided augmentation strategy.
>
>
>
> > ### **W4: Reliance on Confidence Metric Tuning.**
>
> We acknowledge that incorrect confidence estimation may lead to unnecessary context augmentation or premature termination. In our experiments, the introduced confidence metric, combining the weighted KL divergence of the last few layers’ logits with the entropy of the final token distribution, demonstrates robustness and correlates well with the model's inherent capabilities. We observed that weight coefficients tuned on one dataset transfer effectively to other datasets when using the same model.
>
> Across the models and datasets we evaluated, AdaCache maintains stable performance when the KL divergence weight falls within [0.6, 0.8] and the confidence threshold ranges between [0.8, 0.9]. This relatively wide stable range indicates our confidence metric is not overly sensitive to exact hyperparameter values.
>
> > ### **Q1: Performance on larger models.**
>
> We evaluate AdaCache against baselines using the larger Qwen3-14B model on the MMLU, SuperGPQA, and TriviaQA datasets. The results show that AdaCache consistently outperforms Full Recomputation, Prefix Cache, and CacheBlend across all datasets while maintaining comparable generation quality.
>
> | Dataset     | Method          | EM | TTFT (s) |
> |-----------|-----------------|--------|----------|
> | **MMLU** | Full Recompute   |  77.1    | 2.82   |
> |           | Prefix Caching    |  77.1     | 2.47   |
> |           | CacheBlend    |  76.6    |  1.04  |
> |           | **AdaCache (Ours)** | 77.3  | **0.57** |
> | **SuperGPQA** | Full Recompute   |  36.1   |  3.53   |
> |           | Prefix Caching    |   36.1   | 3.12  |
> |           | CacheBlend    |  35.9   | 1.65   |
> |           | **AdaCache (Ours)** |  36.1  | **1.22** |
> | **TriviaQA** | Full Recompute   |  56.5    |  2.87  |
> |           | Prefix Caching    |  56.5  | 2.31  |
> |           | CacheBlend    |  56.0   |  1.15  |
> |           | **AdaCache (Ours)** |  56.3   | **0.72** |

---

### Official Review · Reviewer_1wvZ · 2025-10-31

**Soundness:** 3
**Presentation:** 4
**Contribution:** 3
**Rating:** 6
**Confidence:** 4

**Summary:**

AdaCache proposes a smarter, adaptive framework to speed up RAG in LLMs by solving two common system bottlenecks, repeatedly re-processing the same retrieved context chunks for multiple queries, and indiscriminately adding lots of context even when the query doesn’t need it. To fix this, AdaCache introduces a cache-aware partial recomputation system, profiling the attention flows so it only recalculates what’s actually needed and dynamically matches context prefixes and an adaptive context augmentation strategy based on confidence scores for each query, so simpler questions don’t get massive context of just because. In experiments across several standard datasets (MMLU, SuperGPQA, TriviaQA), AdaCache shows big wins in time-to-first-token , achieving up to 5x speedup over some baselines, all while keeping generation quality intact.

**Strengths:**

First to combine chunk-level attention analytics and hierarchical caching with dynamic, query-by-query adaptive retrieval depth. Demonstrates large and consistent speedups (up to 5x TTFT) across varied tasks, while maintaining or improving answer quality. Empirical coverage multiple models, datasets, real hardware setup, system code/pseudocode details included. Addresses two core RAG bottlenecks context reuse, variable context needs, rather than just one. Well-justified caching logic, tested across different chunk retrieval densities.

**Weaknesses:**

Technical sections like attention analysis, selective recomputation ratio could use more intuitive explanation for ML domain readers. Adaptive context augmentation relies on confidence measurements that may need more validation for edge-cases/unusual queries. Tested mainly on general QA/benchmark tasks. broader evaluation with multi-turn dialogue, enterprise data, multilinguality would add a lot. Efficiency gains are only as good as the popularity distribution of text chunks. if access is more uniform, the cache hit rates drop. Hardware setup is fairly high-end. results might differ for lower-resourced deployments.

**Questions:**

Could the adaptive context augmentation framework be extended to multi-hop QA or dialogue, with context dependencies beyond the initial retrieval? How robust is the cache-aware recomputation under rapidly changing corpora e.g., streaming or up-to-date knowledge bases, what is the trade off? Are there edge-case queries where adaptive context addition either under-fetches (missing context) or over-fetches (noise)? Any way you could provide results for wider domain/task evaluation beyond QA/reading comprehension?

---

> ### Author Response · Authors · 2025-11-22
>
> **We sincerely thank the reviewer for the constructive feedback. We appreciate the recognition of our contributions and would like to address the concerns raised:**
>
>
>  > ### **W1: Technical sections use more intuitive explanations.**
>
> We appreciate this suggestion. In the revised manuscript, we will enhance the presentation of attention analysis and selective recomputation mechanisms with more accessible explanations.
>
>  > ### **W2 & Q3: Adaptive context augmentation relies on confidence measurements .**
>
> We acknowledge the reviewer’s concern regarding the reliance on confidence measurements. In our experiments, the designed confidence estimation, which combines weighted KL divergence of the last few layers’ logits with the entropy of the output token distribution, shows robust behavior and aligns well with model capabilities. Importantly, we find that weight coefficients tuned on one dataset transfer effectively to others when using the same model.
>
> Across all evaluated models and datasets, AdaCache maintains stable performance when the KL divergence weight lies within [0.6, 0.8] and the confidence threshold ranges between [0.8, 0.9]. This wide stability range indicates that the confidence estimation works reliably without requiring precise tuning for different queries or edge cases.
>
>  > ### **W3 & Q1: Broader evaluation on multi-hop QA task.**
>
> We conducted experiments on two popular multi-hop QA datasets using Qwen3-4B and Qwen3-8B. These datasets require synthesizing information across multiple retrieved documents to answer complex queries. The results demonstrate that AdaCache consistently outperforms Full Recomputation, Prefix Cache, and CacheBlend across all datasets and models while maintaining comparable generation quality. These validate AdaCache's effectiveness on tasks requiring complex reasoning.
>
> | Dataset | Model     | Method          | EM | TTFT (s) |
> | ----------- |-----------|-----------------|--------|----------|
> |  **2WikiMultihopQA** | **Qwen3-4B** | Full Recompute   | 42.1      |  1.31   |
> | |           | Prefix Caching    | 42.1      |   1.16   |
> | |           | CacheBlend        | 41.5      |  0.74   |
> | |           | **AdaCache (Ours)** | 41.9    | **0.63** |
> | | **Qwen3-8B** | Full Recompute   | 45.4      |  1.84   |
> | |           | Prefix Caching    | 45.4      |  1.58  |
> | |           | CacheBlend        | 45.3      |  1.04   |
> | |           | **AdaCache (Ours)** | 45.6    | **0.76** |
> | **HotpotQA** | **Qwen3-4B** | Full Recompute   | 40.3      | 1.19    |
> | |           | Prefix Caching    | 40.3      | 1.11     |
> | |           | CacheBlend        | 39.8      | 0.73     |
> | |           | **AdaCache (Ours)** | 40.1    | **0.68** |
> | | **Qwen3-8B** | Full Recompute   | 42.7      | 1.86     |
> | |           | Prefix Caching    | 42.7      | 1.73     |
> | |           | CacheBlend        | 42.4      | 0.99    |
> | |           | **AdaCache (Ours)** | 42.5    | **0.88** |
>
>
>  > ### **W4: Impact of chunk popularity distribution.**
>
> We acknowledge that cache hit rates may depend on the popularity distribution of retrieved chunks. In practice, power-law distribution is common in real-world QA applications. As shown in Fig. 1 of the manuscript, we observe clear power-law distributions in chunk access patterns on MMLU, TriviaQA, and NaturalQuestions-Open datasets. Notably, on MMLU with top-1 retrieval, the most frequently accessed 10% of text chunks satisfy 80% of all queries.
>
>
>  > ### **W5: Hardware setup and lower-resourced deployments.**
>
> We respectfully note that the NVIDIA RTX A6000 (48GB) used in our evaluation is a mid-tier GPU option, with rental costs substantially lower than those of the A100 and H100. We are currently extending our evaluation to more low-end GPUs to evaluate deployment feasibility better.
>
>  > ### **Q2: Robustness under changing corpora.**
>
> AdaCache is fundamentally corpus-agnostic and integrates seamlessly into existing RAG pipelines without requiring corpus-specific tuning. The effectiveness of cache-aware recomputation stems from the inherent sparsity of attention patterns. First, within each text chunk, only a subset of tokens exhibit significant cross-chunk attention dependencies, while the majority can safely reuse precomputed KV caches. Second, tokens requiring cross-chunk attention typically focus on a limited number of prefix chunks rather than the entire prefix context. Furthermore, our multi-hop QA experiments utilize contexts from different retrieval corpora, validating AdaCache's robustness across diverse knowledge sources.

---

### Official Review · Reviewer_NEXz · 2025-11-02

**Soundness:** 3
**Presentation:** 3
**Contribution:** 3
**Rating:** 6
**Confidence:** 3

**Summary:**

The paper proposes AdaCache, a RAG-oriented serving framework that (i) builds a 3-tier, attention-aware KV cache (hard prefix, soft prefix, independent) and selectively recomputes only attention-critical tokens, and (ii) runs an adaptive context augmentation loop that stops adding retrieved chunks once the model is confident. On QA-style RAG benchmarks and several open LLMs, it reports 1.4×–5× TTFT improvements over prior caching systems while keeping accuracy.

**Strengths:**

- Tackles an actually painful RAG-serving bottleneck and is compatible with existing inference stacks.

- Sensible combination of attention-guided partial recomputation with incremental context — the two parts reinforce each other.

- Empirical results show clear win over prefix-only caching and even over CacheBlend in long-context regimes.

**Weaknesses:**

- Each core idea has close prior art: RAGCache / PromptCache / CacheBlend on the caching side, and Active / Adaptive / Bandit RAG on the context-length side; the paper currently oversells novelty.

- Experiments do not ablate how much gain comes from "adaptive context" vs "better cache reuse", and they do not compare to adaptive-retrieval baselines.

- The proposed confidence computation and attention-sink profiling may add non-trivial overhead and may not generalize to other models/tasks.

**Questions:**

- Can you report an ablation that fixes the cache to CacheBlend-style selective recompute and only adds your ACA loop, to show the incremental benefit?

- How exactly is the soft-prefix cache key defined, and what prevents explosion in cache entries when the sink chunk moves?

- What is the actual extra latency of computing per-layer logits for the confidence score on your hardware, and is it still a win when retrieval is remote / high-latency?

---

> ### Author Response · Authors · 2025-11-21
>
> **We sincerely thank the reviewer for the constructive feedback. We appreciate the recognition of our contributions and would like to address the concerns raised:**
>
> ### **W1: Difference from prior caching and adaptive RAG methods.**
>
> **On the caching side**: AdaCache introduces a hierarchical caching architecture that fundamentally differs from prior work by bridging the gap between strict prefix matching and fully independent caching. RAGCache requires exact prefix matching, and CacheBlend adopts independent chunk caching with uniform recomputation strategies, ignoring the attention patterns across chunks. We designed soft prefix caching. By matching only the effective prefix (predecessor or sink chunk), soft prefix cache enables flexible reuse while requiring lower recomputation ratios than CacheBlend, since tokens only need partial attention restoration rather than full cross-chunk attention reconstruction.
>
> **On the context-length side**: While methods like Adaptive RAG determine whether to retrieve, AdaCache determines the precise retrieval depth (exact top-k) for each query. The efficiency gain comes from tight integration with caching module. Each context augmentation step incurs minimal overhead since previously processed chunks achieve hard prefix cache hits without recomputation.
>
> ### **W2 & Q1: Ablation study demonstrating incremental benefits.**
>
> We provide the ablation comparing CacheBlend-style recomputation with and without Adaptive Context Augmentation (ACA) using Qwen3-8B model on different datasets:
>
>
> | Dataset     | Method          | EM | TTFT (s) |
> |-----------|-----------------|--------|----------|
> | **MMLU** | CacheBlend  | 76.1      |  0.77   |
> |           | CacheBlend + ACA    |   76.5    |  0.51  |
> |           | **AdaCache (Ours)** |  76.7   | **0.42** |
> | **SuperGPQA** | CacheBlend   |   33.4   |  0.83   |
> |           | CacheBlend + ACA     |   33.7   |  0.79  |
> |           | **AdaCache (Ours)** |  33.8   | **0.72** |
> | **TriviaQA** | CacheBlend   |   43.7    |   0.72  |
> |           | CacheBlend + ACA     |  44.0    | 0.64   |
> |           | **AdaCache (Ours)** |  43.9  | **0.49** |
>
> Integrating ACA with CacheBlend yields speedups of 1.51×, 1.06×, and 1.13× on MMLU, SuperGPQA, and TriviaQA, respectively. The substantial gains on MMLU stem from its query distribution — most of questions can be correctly answered with minimal context, allowing ACA to terminate context augmentation early. Conversely, SuperGPQA and TriviaQA contain a higher proportion of unanswerable queries (i.e., the model can only achieve approximately 40% accuracy) that force full context expansion, limiting ACA's gains.
>
> ### **W3 & Q3: Regarding confidence computation and attention-sink profiling.**
> **Confidence computation overhead**: The overhead is negligible (<1% of prefill overhead). AdaCache does not compute logits for every layer. We only compute logits for the last 3–4 layers to measure the layer-consistency KL divergence. Moreover, logits are computed only for the last token, not the entire context.
>
> **Attention-sink profiling**: We validated chunk-level attention patterns across Llama3-8B-Instruct, Qwen3-4B, and Qwen3-8B models on MMLU, MMLU-Pro, TriviaQA, and SuperGPQA datasets. We observed consistent behaviors similar to those shown in Fig. 4 of our manuscript, though the specific chunk positions serving as attention sinks vary across different contexts.
>
> **High-latency retrieval**: Even with high-latency remote retrieval, AdaCache outperforms baselines because retrieval cost is identical across methods, and AdaCache reduces the dominant prefill bottleneck. ACA does not increase retrieval overhead—top-k retrieval is executed once per query, and then ACA incrementally incorporates the already-retrieved chunks into the prompt.
>
> We have added these clarifications in Section 3.2 of the revised manuscript.
>
> ### **Q2: Soft prefix cache key definition.**
>
> The chunk-level attention patterns shown in Fig. 4 of the manuscript reveal a distinct two-phase behavior across model depth. In the first half of layers, chunks predominantly focus on their immediate predecessors. In the second half, an attention sink phenomenon emerges. Hence, we set $Cache key = hash([predecessor\\_chunk, current\\_chunk])$ for the first half of layers, and set $Cache key = hash([sink\\_chunk, predecessor\\_chunk, current\\_chunk])$ for the second half of layers. A text chunk can be retrieved from soft prefix cache space only when its effective prefix (predecessor or sink) chunk matches.
>
> Although predecessor/sink chunks may vary, this does not cause exponential growth of cache entries. In fact, the combinations also follow a power-law distribution, similar to Fig. 2a in the manuscript. For top-2 retrieval on MMLU, 29% of joint chunks are retrieved by 80% of queries.

---

> ### Author Response · Authors · 2025-12-03
> **Additional experimental results**
>
> ### **W2 & Q1: Ablation study demonstrating incremental benefits.**
>
> We further conduct performance (TTFT) comparisons where the baselines are combined with adaptive context augmentation (ACA), while ensuring that all evaluations are conducted under comparable generation quality. ACA integrates smoothly with different caching mechanisms, and when applied to Prefix Caching or CacheBlend, it consistently reduces TTFT across models and datasets, yielding average speedups of 1.65× and 1.22×, respectively.
>
> The full AdaCache system maintains clear performance advantages over these ACA-enhanced baselines, achieving average speedups of 1.76× over Prefix Caching and 1.23× over CacheBlend. Notably, cache-aware selective recomputation alone can occasionally outperform even the ACA-enhanced baselines. These results highlight the effectiveness of our hierarchical caching design and its close synergy with ACA.
>
> We have added these additional ablation studies in Section 4.2 of the revised manuscript.
>
>
> | Model      | Dataset     | Prefix Caching | Prefix Caching + ACA | CacheBlend | CacheBlend + ACA | AdaCache w/o ACA | AdaCache w/ ACA |
> |------------|-------------|----------------|-----------------------|------------|-------------------|-------------------|----------|
> | **Qwen3-4B** | MMLU        | 1.00 | 0.41 | 0.48 | 0.29 | 0.38 | 0.20 |
> |            | MMLU-Pro    | 0.94 | 0.70 | 0.68 | 0.63 | 0.61 | 0.56 |
> |            | SuperGPQA   | 1.06 | 0.75 | 0.49 | 0.46 | 0.47 | 0.44 |
> |            | TriviaQA    | 0.80 | 0.48 | 0.43 | 0.34 | 0.36 | 0.22 |
> | **Qwen3-8B** | MMLU        | 1.56 | 0.72 | 0.77 | 0.51 | 0.62 | 0.42 |
> |            | MMLU-Pro    | 1.70 | 1.18 | 0.93 | 0.88 | 0.90 | 0.82 |
> |            | SuperGPQA   | 1.65 | 1.33 | 0.83 | 0.79 | 0.85 | 0.72 |
> |            | TriviaQA    | 1.42 | 0.96 | 0.72 | 0.64 | 0.59 | 0.49 |

---

### Author Response · Authors · 2025-12-03
**Summary Comment**

We thank the Area Chair, Senior Area Chair, and Program Chair for your efforts in coordinating the review process, and we appreciate all reviewers for their constructive comments. Below, we summarize the strengths recognized by reviewers, as well as the main concerns and our responses.

---

## **Strengths Recognized by Reviewers**
- Tackles real RAG serving bottlenecks with strong practical motivation (`NEXz` , `1wvZ` , `DScb` , `8B2T`)
- Achieves substantial performance improvements over prior caching systems while maintaining accuracy (`NEXz` , `1wvZ` , `DScb` , `8B2T`)
- Sensible combination of attention-guided partial recomputation with incremental context augmentation (`NEXz` , `1wvZ` , `DScb`)
- Strong adaptability and broad applicability across diverse datasets and LLMs (`1wvZ` , `DScb` , `8B2T`)
- Well-justified and thoughtfully designed caching architecture (`1wvZ` , `DScb`)

---

## **Main Concerns & Our Responses**

**1. Difference from Prior Caching and Adaptive RAG Methods**

**Our Response:** AdaCache introduces two key innovations:
- **Hierarchical caching:** Bridges strict prefix matching and independent caching through soft prefix cache that matches only effective prefixes, improving cache hit rate and minimizing token recomputation.
- **Adaptive retrieval depth:** Unlike prior work that decides *whether* to retrieve, AdaCache determines the precise top-k for each query, integrating tightly with cache reuse.

**2. Ablation Studies**

**Our Response:** We conducted comprehensive ablation studies and included them in Section 4.2 of the revised manuscript. The results demonstrate the individual effectiveness of both the hierarchical caching mechanism and the adaptive context augmentation strategy.


**3. Confidence Computation and Extra Retrieval Overhead**

**Our Response:** We clarified that confidence estimation introduces less than 1% additional prefill cost and does not incur any extra retrieval overhead.

**4. Attention Pattern Consistency**

**Our Response:** We validated chunk-level attention patterns across Llama3-8B-Instruct, Qwen3-4B, and Qwen3-8B on MMLU, MMLU-Pro, TriviaQA, and SuperGPQA, observing consistent behaviors.

**5. Multi-hop QA and Complex Scenarios**

**Our Response:** We conducted experiments on two multi-hop QA datasets (2WikiMultihopQA and HotpotQA). Results demonstrate AdaCache consistently outperforms all baselines. These have been added to Section 4.2, Figure 5 in revised manuscript.

**6. Chunk Popularity Distribution Impact**

**Our Response:** We clarified that the observed power-law distribution is common in real-world QA applications and is validated across multiple datasets.

**7. Comparison with TurboRAG**

**Our Response:** TurboRAG requires costly model retraining using 32 A100 GPUs. AdaCache is training-free and introduces minimal overhead. Moreover, our adaptive context augmentation is orthogonal to TurboRAG's caching strategy, combining both could further reduce KV-cache loading costs.

**8. Applicability beyond Open-weight Models**

**Our Response:** We clarified that AdaCache is not limited to open-weight models. AdaCache is a system-level optimization that LLM vendors can incorporate without exposing model weights.

---

Although we have not yet received any follow-up from reviewers, we believe all concerns have been effectively addressed.

---

### Meta-Review · Area_Chair_kDVM · 2025-12-23

**Summary:**

### Summary
The paper proposes AdaCache, a RAG-oriented serving framework that (i) builds a 3-tier, attention-aware KV cache (hard prefix, soft prefix, independent) and selectively recomputes only attention-critical tokens, and (ii) runs an adaptive context augmentation loop that stops adding retrieved chunks once the model is confident.

### Reviewer summary

Strengths: reviewers generally find the work targets real serving bottlenecks and reports strong TTFT gains with clear engineering value.

Key concerns: (i) evaluation is mostly QA/RAG (initially close to single-chunk QA), with limited evidence for broader serving workloads; (ii) novelty/positioning vs related work may be overstated; (iii) deployment assumptions are strong (KV-cache/attention control), raising reproducibility concerns in closed stacks.

### AC Comments
The rebuttal adds key ablations to disentangle ACA vs caching effects, includes multi-hop QA results, and clarifies ACA overhead—materially addressing the main insufficient evidence critiques. Remaining weaknesses are mainly around novelty framing and applicability boundaries (broader workloads and deployability in closed inference stacks). The final version should further tighten claims and clearly state assumptions/limitations.

**Reviewer Concerns:**

### Reviewer NEXz

1. Resolved

clearer ablations to separate ACA vs caching/recompute contributions.

2. outstanding

novelty: Author argued hierarchical caching (hard/soft/independent) bridges strict prefix matching and independent chunk caching, and proposed “exact top-k adaptive injection” tightly coupled with caching; explained soft-prefix keys and cache blow-up avoidance. The response is mainly positioning; it still lacks more systematic, apples-to-apples comparisons and a tighter summary of what is truly new. Claims should be further tightened.


### Reviewer 1wvZ

Resolved:

* evaluation too simple; requested more complex (multi-hop) QA.

* coverage is still narrow beyond QA

### Reviewer DScb

Resolved:

* ACA overhead and whether it is cost-effective (including remote retrieval).

* validation on a larger model.



### Reviewer 8B2T

1. Resolved

* evaluation initially looked like “single-fact QA”; requested harder tasks.


2. Outstanding

deployability/reproducibility—requires KV-cache/attention control, which is difficult for third parties / closed inference stacks.

Author argued system optimizations can be integrated inside inference frameworks or by closed vendors without exposing weights.
But, “vendors can integrate” does not imply community reproducibility; assumptions and deployment modes should be explicitly stated to avoid implying general applicability.

**Reviewer Scores:**

1. NEXz (6): likely unchanged after the new ablations.

2. 1wvZ (6): likely unchanged; multi-hop results strengthen the case.

3. DScb (6): likely unchanged after overhead + large-model evidence.

4. 8B2T (4): could move to 6, but may remain cautious due to deployability assumptions.

---

### Decision · Program_Chairs · 2026-01-26

Accept (Poster)